# Pregnant women's migration patterns before childbirth after large-scale earthquakes and the added impact of concerns regarding radiation exposure in Fukushima and five prefectures

**Yuta Inoue**[1¤]*, **Kazutomo Ohashi**[2☉], **Yuko Ohno**[3☉], **Takako Fujimaki**[3☉], **Anna Tsutsui**[3☉], **Ling Zha**[1☉], **Tomotaka Sobue**[1☉]

1 Division of Environmental Medicine and Population Sciences, Graduate School of Medicine Osaka University, Osaka, Japan, 2 Faculty of Global Nursing, Otemae University, Osaka, Japan, 3 Division of Health Sciences, Graduate School of Medicine, Osaka University, Osaka, Japan

☉ These authors contributed equally to this work.
¤ Current address: Department of Medical Treatment Recover Care Nursing, Graduate School of Biomedical Sciences, Tokushima University, Tokushima, Japan
* inoue@envi.med.osaka-u.ac.jp

## Abstract

The 2011 Great East Japan Earthquake (within Fukushima, Iwate, and Miyagi prefectures) was a complex disaster; it caused a tsunami and the Fukushima Daiichi Nuclear Power Plant accident, resulting in radiation exposure. This study investigated the earthquake's effects on the migration patterns of pregnant women and their concerns regarding radiation exposure. We also considered the following large-scale earthquakes without radiation exposure: Great Hanshin-Awaji (Hyogo prefecture), Niigata-Chuetsu, and Kumamoto. Pregnant women were categorized as outflow and inflow pregnant women. Data on the annual number of births three years before and after the earthquake were used as a denominator to calculate the outflow and inflow rates per 100 births. The odds ratios of annual outflow and inflow rates after the earthquake, using three years before the earthquake as the baseline, were calculated. The odds-ratio for outflow significantly increased for Hyogo, Fukushima, Miyagi, and Kumamoto prefectures after the earthquake, particularly for Fukushima, showing a significant increase until three years post the Great East Japan Earthquake (disaster year: odds-ratio: 2.66 [95% confidence interval: 2.44–2.90], 1 year post: 1.37 [1.23–1.52], 2 years post: 1.13 [1.00–1.26], 3 years post: 1.18 [1.05–1.31]), while the remaining three prefectures reported limited increases post one year. The inflow decreased after the earthquake, particularly in Fukushima, showing a significant decrease until 2 years post the Great East Japan Earthquake (disaster year: 0.58 [0.53–0.63], 1 year post: 0.76 [0.71–0.82], 2 years post: 0.83 [0.77–0.89]). Thus, pregnant women's migration patterns changed after large-scale earthquakes, suggesting radiation exposure concerns possibly have a significant effects. These results suggested that plans for receiving assistance and support that considers the peculiarities of disaster related

**Data Availability Statement:** Data that we used cannot be shared publicly for participant privacy in accordance with Statistics Act in Japan. Based on the act, data are available from the Ministry of Health, Labor and Welfare. Institutional Data Access/ Ethics Committee (contact via Ministry of Health, Labor and Welfare) for researchers who officially request to data. Tel: +81-3-5253-1111. https://www.mhlw.go.jp/toukei/sonota/chousahyo. html.

**Funding:** This work was supported by Research project on the Health Effects of Radiation organized by Ministry of the Environment, Japan. The funders had no role in study design, data collection and analysis, decision to publish, or preparation of the manuscript.

**Competing interests:** The authors have declared that no competing interests exist.

damage and pregnant women's migration patterns are needed in both the affected and non-affected areas.

## Introduction

Japan is an earthquake-prone country. Large-scale earthquakes cause physical damage such as building collapses and disconnection of utilities, like electricity, gas, and water. Earthquakes have a significant impact on the affected population, including changes to the living environment and disruptions to daily functions due to evacuations, resulting in physical and mental burdens [1, 2]. Previous studies on pregnant women affected by disasters have shown that they experience increased anxiety about child-rearing and childbirth, hypertension in pregnancy, gestational diabetes, and other psychological and physical issues [3, 4]. Studies focusing on fetal growth have also shown that experiencing such disasters leads to adverse birth outcomes such as preterm birth and low birth weight [5]. Additionally, earthquakes affect medical institutions and threaten medical care continuity [6–9]. In particular, pregnant women face unique burdens with respect to the continuation of their pregnancies in affected areas, such as difficulties in receiving prenatal checkups due to hindered access to hospitals and medical facilities during childbirth [10]. While an earthquake by itself causes the aforementioned difficulties, the Great East Japan Earthquake (GEJE) of 2011 caused even greater damage due to the secondary tsunami triggered by it [11]. The tsunami caused the Fukushima Daiichi Power Plant accident. The mechanism of adverse health effects caused by the radiation has not yet been clarified, and opinions are divided among experts, which possibly cause anxiety not only among residents of the affected areas but also among the medical members who support them, and may even lead to harmful rumors about the affected areas [12, 13]. In that time, Fukushima Prefecture had conducted The Fukushima Health Management Survey to ascertain the health status of prefectural residents to prevent, detect, and treat diseases at an early stage, and to maintain and promote the health of prefectural residents for the future. Specifically, the pregnancy and birth survey targeted women who received a Maternal and Child Health Handbook between August 1, 2010 and July 31, 2011 from a municipality in Fukushima Prefecture and also the women who received a Maternal and Child Health Handbook from a municipality outside of Fukushima Prefecture within the same period, but who received a prenatal checkup or gave birth after relocating or returning to Fukushima on or after March 11, 2011 [14, 15]. Results of this survey showed that pregnant women who had conceived and given birth in Fukushima Prefecture were greatly concerned about the effects of radiation on their fetuses and children [16]. Pregnant women not only in Fukushima, but also in Iwate and Miyagi, which were affected areas, had similar concerns [17]. There have been global reports regarding residential mobility during pregnancy in normal times, suggesting that pregnant women are cautious about the effects of various exposures to their fetuses [18–20]. In Japan, there is a unique custom of mobility of pregnant women called "Satogaeri" in normal times [21]. According to a study, after earthquakes, pregnant women decided against giving birth at their chosen facilities and gave up the customary Japanese practice of returning to their parents' home to give birth [17]. Thus, we consider that during a disaster, there will be more outflow of pregnant women from the affected areas to non-affected areas and a fewer inflow of pregnant women from non-affected areas to affected areas. Pregnant women's migration patterns can be investigated using address information recorded in the Vital Statistics birth registry, which we obtained the birth registry data from the Ministry of Health, Labour and Welfare. The purpose of this study is to understand the actual migration patterns, from the outflow and inflow rates of

pregnant women based on the number of births before and after large-scale earthquakes using the Vital Statistics birth registry. It explores the relationship between the degree of damage and pregnant women's migration patterns by comparing data pertaining to past large-scale earthquakes. It also clarifies that even for earthquakes where no added concerns for radiation exposure were present, changes in the migration patterns of pregnant women after the event was related to the magnitude of the effect of the disaster.

## Methods

### Definition of complex disaster and targeted large-scale earthquake

In this study, earthquakes with radiation concerns were treated as complex disasters. Therefore, among the prefectures of Iwate, Miyagi, and Fukushima that were affected by the GEJE, only Fukushima Prefecture was treated as having been affected by the complex disaster, because concerns regarding radiation exposure were high and widespread in Fukushima, whereas this was true only for a few areas in Miyagi and none in Iwate [22]. Large-scale earthquakes were defined as earthquakes with a magnitude greater than 6.5 and a seismic intensity scale of more than 7 that occurred after 1995, including the Great Hanshin-Awaji Earthquake (GHAE, 1995), Niigata Chuetsu earthquake (2004), Kumamoto earthquake (2016), and GEJE (2011) [23, 24].

Table 1 shows the damage caused by past large-scale earthquakes. In terms of human casualties and damaged houses, Hyogo Prefecture, which was hit by the GHAE, and Miyagi Prefecture, which was hit by the GEJE, suffered the most damage. However, it is not possible to simply compare the number of people affected because it is necessary to consider the age

**Table 1. Details of the damage caused by large-scale earthquakes.**

| Prefecture | Fukushima[d] | Iwate[f] | Miyagi[g] | Hyogo[h] | Niigata[j] | Kumamoto[k] |
|---|---|---|---|---|---|---|
| **Erathquake** | The Great East Japan Earthquake | | | The Great Hanshin-Awaji Earthquake | The Niigata Chuetsu Earthquake | The Kumamoto Earthquake |
| **Date of disaster** | 3, 11, 2011 | | | 1, 17, 1995 | 10, 23, 2004 | 4, 14 and 16, 2016 |
| **Population density[a,l] (persons/km$^2$)** | 144.2 | 86.1 | 319.3 | 644.1 | 194.3 | 242.3 |
| **Magnitude of earthquake** | M9.0 | M9.0 | M9.0 | M7.3 | M6.8 | M6.5 and 7.3 |
| **Human casualties** | | | | | | |
| Number of dead or unknown | 4,162 | 5,823 | 11,785 | 6,437 | 68 | 216 |
| Number of Injured | 183 | 206 | 4,117 | 43,792 | 4,795 | 2,673 |
| **Number of houses damaged[b]** | 98,218 | 24,916 | 238,135 | 249,180 | 16,985 | 42,192 |
| **Number of evacuees[c]** | No data | 54,429 | 320,885 | 307,022[i] | 103,000 | 183,882 |
| **Number of power outage[c]** | 374,989[e] | 760,000 | 1.54 million | 2.6 million | 300,000 | 470,000 |
| **Number of gas outage[c]** | 16,998[e] | 9,400 | No data | 860,000 | 56,000 | 105,000 |
| **Number of water outage[c]** | No data | 180,000 | No data | 1.3 million | 130,000 | 445,857 |
| **Radiation exposure concern** | Yes | No | No | No | No | No |

[a] Population density is shown for the year of the earthquake.

[b] Including not only fully destroyed but also partially destroyed houses.

[c] Due to different accounting methods for aftershocks, the number of reports may not match in the literature. Regarding damage to utilities and the number of people affected by the disaster, only prefectures that reported the maximum values are listed because these data fluctuated with the passage of days after the disaster.

[d–l] Reference number; For the annotations d to l, please see references [25–33], respectively.

distribution of resident populations as well as the population density. Furthermore, Japan is prone to disasters such as typhoons and earthquakes and the Building Standard Law has been revised many times, each time increasing the strength of buildings. Therefore, the seismic resistance of houses and buildings were different over time.

## Data sources

For this study, we used the information on residential and notified addresses recorded by the Vital Statistics birth registry. Japan's Vital Statistics is as accurate and complete as possible because every birth must be registered by law. The registry reports the residential address of the newborn but not that of the mother. However, in this study, we considered the residence of the child as the residence of the mother, because the newborn child needs to be cared for by the guardian. The address of the notified place refers to the prefecture of birth or location of the notifier, and it includes the area where the child was born and where the child temporarily lives, including evacuation areas. The analysis period is three years before and after the date of each earthquake. To use these data of individuals, we applied for and obtained data for 1992 to 2019 from the Ministry of Health, Labour and Welfare. Of the 31,020,751 individual birth reports in the registry for the 1992–2019 period, those with unknown or missing information such as residence address, notification address, maternal age, or number of weeks pregnant were excluded. As a result, the final analysis included 30,311,671 cases.

## Calculating the rate of flow of pregnant women

We used two sets of information from the birth registry: the residence address (where the child was registered as a resident) and birth notification address (where the child was born). Based on this information, when a pregnant woman residing in a prefecture that was close to the epicenter of an earthquake (hereinafter referred to as the "severely affected prefecture") moved to another prefecture to give birth, we defined it as outflow; when a pregnant woman moved to the severely affected prefecture from a different prefecture to give birth, it was defined as inflow. For calculating the rate of outflow of pregnant women, the denominator is the number of births and the place of residence was in the one severely affected prefecture. The numerator is the number of births for which the place of residence was in the one severely affected prefecture and for which the place of notification was in the 46 prefectures except for their place of residence. For calculating the rate of inflow of pregnant women, the denominator is the number of births and the place of residence was in 46 prefectures except for the one severely affected prefecture. The numerator is the number of births for which the place of residence was in 46 prefectures except for the one severely affected prefecture and for which the place of notification was in the one severely affected prefecture. It is possible that the areas with government restrictions for returning are not areas where pregnant women are willing to move, but areas where they are ordered to move by the government, and the background that determined their movement is different. However, the Vital Statistics birth registry used in this study could not grasp these backgrounds. Therefore, in this study, the 13 areas ordered for evacuation were also treated as residential areas of pregnant women, without distinguishing between the intention of pregnant women and administrative restrictions on return. To keep the number of days elapsed from the date of each earthquake the same, we counted the annual data, starting from the date of occurrence of each earthquake, instead of collecting them by the calendar period. The year when the earthquake occurred was defined as the disaster year starting from the date of the disaster, followed by one year post, two years post, and three years post. Similarly, the time before the disaster year was defined as one year ago, followed by two and three years ago.

## Methods of calculating the ORs for the outflow and inflow rates

The baseline was the three-year period before the earthquake from three years to one year ago, and the odds ratios (ORs) for each of the three years since the earthquake were calculated using logistic regression analysis with migration patterns as the objective variable and the year since the earthquake, maternal age, and number of children as explanatory variables. Maternal age was classified into seven age groups ($\leq$ 19, 20–24, 25–29, 30–34, 35–39, 40–44, $\geq$ 45 years), and the number of children was classified into one, two, and three or more. A p-value $< 0.05$ was considered statistically significant. Statistical analyses were performed using JMP pro 15.

## Ethical approval

This study was conducted with the approval of the Research Ethics Committee of Ethical Review Board of Osaka University Hospital (Approval number: 15272–6). Patent consent was waived because this retrospective case analysis involved de-identified data for research purposes in accordance with the Statistics Act of Japan.

## Results

### Annual trends of the outflow of pregnant women and impact of large-scale earthquakes

The outflow rates of pregnant women are shown in Fig 1. The outflow rate in Hyogo and Miyagi Prefectures was higher than that in the other prefectures throughout the entire period (three years before and after the year of the earthquake, Hyogo: 7.61%–6.32%, Miyagi: 5.08–5.83%). Table 2 shows the ORs of the outflow rates of pregnant women for the years since the earthquake, maternal age, and number of children. For Hyogo, Fukushima, Miyagi, and Kumamoto, the outflow of pregnant women from the severely affected prefecture in the disaster year showed a significant increase compared to the baseline. Particularly in Fukushima, where there was a specific concern about radiation exposure, the outflow in the year of the earthquake was significantly higher than in the other prefectures (ORs for the disaster year: 2.66, confidence interval [CI]: 2.44–2.90) and it significantly increased until three years after the earthquake (ORs for 1 year post: 1.37, CI: 1.23–1.52; ORs for post 2 year post: 1.12, CI:

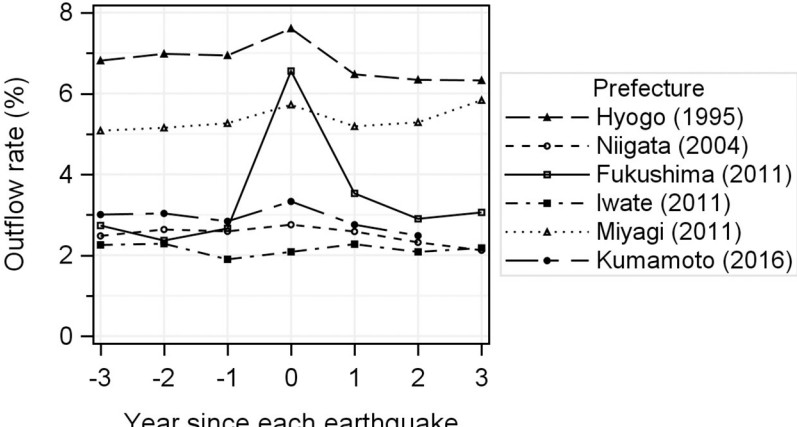

**Fig 1. Change in outflow rate in the three years before and after the year of the earthquake (per 100 births).**

**Table 2. Odds ratios of outflow of pregnant women in the post-disaster period.**

| | Hyogo(1995) | Niigata(2004) | Fukushima(2011) | Miyagi(2011) | Iwate(2011) | Kumamoto (2016) |
|---|---|---|---|---|---|---|
| Variable | ORs [a] [95% CI[b]] | ORs [a] [95% CI] | ORs [a] [95% CI] | ORs [a] [95% CI] | ORs [a] [95% CI] | ORs [a] [95% CI] |
| **Year since each earthquake[d]** | | | | | | |
| **Baseline** | | | | | | |
| (-1,-2,-3) | Ref. [c] | Ref. [c] | Ref. [c] | Ref. [c] | Ref. [c] | Ref. [c] |
| 0 | 1.09 [1.05–1.14] | 1.07 [0.97–1.19] | 2.66 [2.44–2.90] | 1.13 [1.05–1.22] | 0.98 [0.83–1.15] | 1.14 [1.02–1.26] |
| 1 | 0.91 [0.88–0.95] | 1.01 [0.91–1.12] | 1.37 [1.23–1.52] | 1.02 [0.95–1.10] | 1.07 [0.91–1.25] | 0.94 [0.84–1.06] |
| 2 | 0.89 [0.85–0.92] | 0.90 [0.81–1.01] | 1.12 [1.00–1.26] | 1.04 [0.96–1.12] | 0.74 [0.83–1.15] | 0.85 [0.76–0.96] |
| 3 | 0.88 [0.84–0.91] | 0.83 [0.75–0.93] | 1.18 [1.05–1.31] | 1.15 [1.07–1.24] | 1.01 [0.86–1.19] | |
| **Maternal age (years)** | | | | | | |
| ≤ 19 | 0.20 [0.17–0.23] | 0.24 [0.15–0.38] | 0.42 [0.31–0.57] | 0.14 [0.09–0.20] | 0.21 [0.11–0.43] | 0.14 [0.07–0.29] |
| 20–24 | 0.52 [0.50–0.54] | 0.52 [0.47–0.58] | 0.61 [0.55–0.67] | 0.53 [0.49–0.57] | 0.47 [0.40–0.56] | 0.50 [0.43–0.59] |
| 25–29 | Ref. [c] | Ref. [c] | Ref. [c] | Ref. [c] | Ref. [c] | Ref. [c] |
| 30–34 | 1.10 [1.07–1.12] | 1.04 [0.98–1.11] | 1.11 [1.04–1.19] | 0.96 [0.92–1.01] | 1.06 [0.96–1.18] | 1.08 [0.99–1.19] |
| 35–39 | 0.75 [0.72–0.79] | 0.78 [0.70–0.86] | 1.06 [0.97–1.15] | 0.70 [0.65–0.74] | 0.97 [0.85–1.10] | 0.74 [0.65–0.83] |
| 40–44 | 0.47 [0.40–0.56] | 0.63 [0.48–0.82] | 0.89 [0.74–1.07] | 0.45 [0.39–0.52] | 0.61 [0.45–0.83] | 0.56 [0.44–0.72] |
| ≥ 45 | 0.53 [0.19–1.43] | 0.56 [0.08–4.29] | 0.26 [0.04–1.83] | 0.19 [0.05–0.76] | unstable | 0.58 [0.14–2.35] |
| **Number of children** | | | | | | |
| 1 | Ref. [c] | Ref. [c] | Ref. [c] | Ref. [c] | Ref. [c] | Ref. [c] |
| 2 | 0.77 [0.75–0.79] | 0.66 [0.62–0.70] | 0.79 [0.74–0.84] | 0.72 [0.69–0.75] | 0.66 [0.60–0.72] | 0.77 [0.71–0.83] |
| ≥ 3 | 0.25 [0.24–0.27] | 0.19 [0.17–0.22] | 0.42 [0.38–0.46] | 0.22 [0.20–0.24] | 0.22 [0.18–0.26] | 0.19 [0.17–0.23] |

[a] ORs, odds ratios;

[b] CI, confidence interval;

[c] Ref, reference.

[d] Baseline is a period of 3 years before the disaster year. For the number of years since the earthquake, 0 indicates disaster year. For the number of years since the earthquake, 1, 2, 3, -1, -2 and -3 indicate 1 year post, 2 years post, 3 years post, 1 year ago, 2 years ago, and 3 years ago, respectively.

1.00–1.25; and ORs for 3 year post: 1.18, CI: 1.05–1.31), while increases were limited for 1 year post the earthquake year in the remaining three prefectures. The outflow rate of pregnant women aged 25–34 years was higher than that in the other age groups, and younger or older pregnant women had a lower outflow rate. Focusing on the number of children and outflow of pregnant women, we found an inverse correlation.

Kumamoto Prefecture was affected by the earthquake in 2016, and the ORs were calculated only for the 2-years-post period of the earthquake. For the number of years since the earthquake, 0 indicates the disaster year. For the number of years since the earthquake, 1, 2, 3, -1, -2, and -3 indicate 1 year post, 2 years post, 3 years post, 1 year ago, 2 years ago, and 3 years ago, respectively.

## Annual trends of inflow of pregnant women and impact of large-scale earthquakes

The inflow rates of pregnant women are shown in Fig 2. The rate in Hyogo Prefecture was higher than in the other prefectures throughout the period (0.22–0.26%). Iwate Prefecture had the lowest rate compared to the other prefectures in this period (0.06–0.07%). Table 3 shows the ORs of the inflow rates for the years since the earthquake, maternal age, and number of children. For Hyogo, Fukushima, Iwate, Miyagi, and Kumamoto, the inflow of pregnant women from the other prefectures in the disaster year showed a significant decrease compared

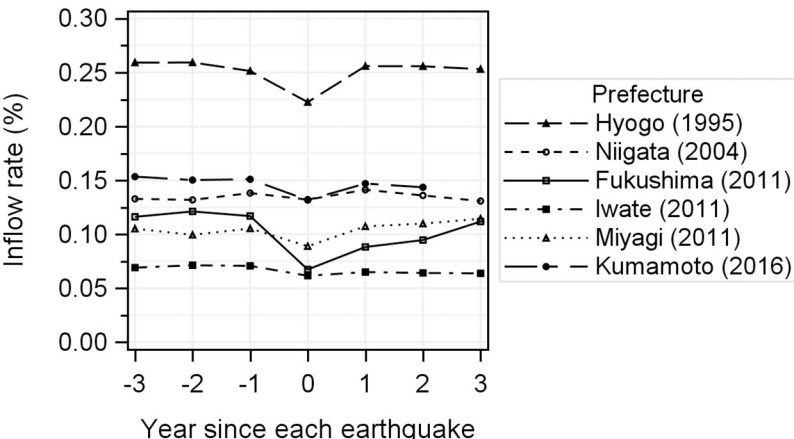

**Fig 2. Change in inflow rate in the three years before and after the year of the earthquake (per 100 births).**

to the baseline. Particularly in Fukushima, with a specific concern about radiation exposure, the inflow in the year of the earthquake was significantly lower than in the other prefectures (ORs for the disaster year: 0.58, CI: 0.53–0.63) and it significantly decreased until 2 years post the earthquake (ORs for 1 year post: 0.76, CI: 0.71–0.82; ORs for 2 years post: 0.83, CI: 0.77–

**Table 3. Odds ratios of inflow of pregnant women in the post-disaster period.**

| | Hyogo(1995) | Niigata(2004) | Fukushima(2011) | Miyagi(2011) | Iwate(2011) | Kumamoto(2016) |
|---|---|---|---|---|---|---|
| Variable | ORs [a] [95% CI] | ORs [a] [95% CI] | ORs [a] [95% CI] | ORs [a] [95% CI] | ORs [a] [95% CI] | ORs [a] [95% CI] |
| **Year since each earthquake[d]** | | | | | | |
| **Baseline** | | | | | | |
| (-1,-2,-3) | Ref. [c] | Ref. [c] | Ref. [c] | Ref. [c] | Ref. [c] | Ref. [c] |
| **0** | 0.86 [0.82–0.89] | 0.99 [0.93–1.05] | 0.58 [0.53–0.63] | 0.87 [0.81–0.94] | 0.89 [0.81–0.97] | 0.87 [0.82–0.93] |
| **1** | 0.98 [0.94–1.02] | 1.07 [1.01–1.05] | 0.77 [0.71–0.82] | 1.05 [0.98–1.13] | 0.95 [0.87–1.03] | 0.98 [0.92–1.04] |
| **2** | 0.97 [0.93–1.01] | 1.04 [0.98–1.10] | 0.83 [0.77–0.89] | 1.08 [1.01–1.16] | 0.95 [0.87–1.03] | 0.95 [0.90–1.02] |
| **3** | 0.95 [0.91–0.99] | 1.01 [0.95–1.07] | 0.98 [0.91–1.05] | 1.12 [1.05–1.20] | 0.94 [0.86–1.03] | |
| **Maternal age (years)** | | | | | | |
| ≤ **19** | 0.16 [0.13–0.20] | 0.12 [0.08–0.16] | 0.15 [0.11–0.21] | 0.14 [0.10–0.20] | 0.13 [0.08–0.20] | 0.41 [0.31–0.53] |
| **20–24** | 0.45 [0.43–0.47] | 0.60 [0.56–0.61] | 0.73 [0.69–0.78] | 0.56 [0.52–0.60] | 0.83 [0.77–0.90] | 0.78 [0.72–0.85] |
| **25–29** | Ref. [c] | Ref. [c] | Ref. [c] | Ref. [c] | Ref. [c] | Ref. [c] |
| **30–34** | 1.13 [1.10–1.16] | 0.93 [0.90–0.97] | 0.81 [0.78–0.85] | 0.93 [0.89–0.97] | 0.75 [0.71–0.79] | 0.85 [0.81–0.90] |
| **35–39** | 0.68 [0.65–0.72] | 0.70 [0.67–0.74] | 0.50 [0.47–0.53] | 0.66 [0.63–0.70] | 0.48 [0.45–0.52] | 0.57 [0.53–0.61] |
| **40–44** | 0.41 [0.34–0.49] | 0.50 [0.43–0.58] | 0.30 [0.26–0.35] | 0.44 [0.39–0.51] | 0.24 [0.19–0.30] | 0.36 [0.31–0.42] |
| ≥ **45** | 0.39 [0.12–1.19] | 0.12 [0.02–0.86] | 0.34 [0.14–0.81] | 0.30 [0.11–0.79] | unstable | 0.28 [0.10–0.74] |
| **Number of children** | | | | | | |
| **1** | Ref. [c] | Ref. [c] | Ref. [c] | Ref. [c] | Ref. [c] | Ref. [c] |
| **2** | 0.93 [0.71–0.74] | 0.67 [0.65–0.70] | 0.75 [0.72–0.78] | 0.68 [0.65–0.70] | 0.79 [0.75–0.83] | 0.90 [0.86–0.94] |
| ≥ **3** | 0.20 [0.19–0.21] | 0.19 [0.17–0.20] | 0.25 [0.22–0.27] | 0.18 [0.16–0.20] | 0.26 [0.23–0.29] | 0.42 [0.38–0.45] |

[a] ORs, odds ratios;

[b] CI, confidence interval;

[c] Ref, reference.

[d] Baseline is a period of 3 years before from the disaster year. For the number of years since the earthquake, 0 indicates disaster year. For the number of years since the earthquake, 1, 2, 3, -1, -2 and -3 indicate 1 year post, 2 years post, 3 years post, 1 year ago, 2 years ago, and 3 years ago, respectively.

0.89), while decreases were limited for a period of 1 year post the earthquake in the remaining four prefectures. An analysis of the relationship among the ages of the pregnant women, the number of existing children, and the inflow rate showed that, similar to the outflow rate, pregnant women aged 25–34 years and primiparas were more likely to migrate.

Kumamoto was affected by the earthquake in 2016, and ORs were calculated only for the 2 years post the earthquake. For the number of years since the earthquake, 0 indicates the disaster year. For the number of years since the earthquake, 1, 2, 3, -1, -2, and -3 indicate 1 year post, 2 years post, 3 years post, 1 year ago, 2 years ago, and 3 years ago, respectively.

## Discussion

### Relationship among large-scale earthquakes and the flow rates of pregnant women

This study revealed that pregnant women moved to other prefectures in the aftermath of the GHAE, Niigata Chuetsu earthquake, GEJE, and Kumamoto earthquake. As shown in Table 1, the results suggest that the human and physical damage caused by large-scale earthquakes as well as the disconnection of utilities, such as electricity, gas, and water affected the migration patterns of pregnant women. This is because if the time taken to restore facilities is long, it may be difficult to secure medical resources and continued medical care, resulting in a temporarily inadequate perinatal care system. It was reported that some pregnant women had to change hospitals due to earthquake damage [34], and some considered moving out of the prefecture. In Fukushima, where concern about radiation exposure led to the disaster being classified as a complex disaster, there was a long-term effect on the migration patterns of pregnant women, and not just in the year of the earthquake. In fact, after the GEJE, pregnant women who lived in Fukushima and the neighboring prefectures reported radiation effects on the fetus and anxiety about breastfeeding [16, 17, 35, 36], and expectant mothers were more worried about childbirth and childcare in Fukushima than in other places after large-scale earthquakes. Radiation cannot be seen or felt, and exposure is known to have a significant effect on fetal growth [37, 38]. Physical distance can prevent radiation exposure, and even 10 years after the GEJE, some zones in Fukushima Prefecture remain under exclusion, forcing residents to live elsewhere [39, 40]. Therefore, radiation exposure can have long-term effects on women planning to have children. As shown above, the migration patterns of pregnant women after large-scale earthquakes may exhibit significant changes that cannot be generally explained by the degree of damage alone when large-scale earthquakes occur in succession, or when a complex disaster that gives rise to concerns about radiation exposure occurs, as in the case of Kumamoto and Fukushima Prefectures. In particular, concerns about radiation exposure have had long-term effects on women's migration patterns.

Additionally, Japan has a custom called "homecoming birth," which has been conceived to reduce the burden of housework and childcare before and after childbirth and help postpartum women achieve mental stability and physical recovery. It is a cultural practice in which the pregnant woman's parents help with all the general household work or take care of older children, so that the mother can get rest [41–43]. Nevertheless, during the GEJE, it was reported that some pregnant women from other prefectures gave up the tradition of homecoming birth [17], which may be associated with the decreased inflow rate from a different prefecture to give birth. Thus, we believe that the outflow and inflow rates shown in this study are indicative of the overall evacuation behavior of pregnant women, including evacuation to safe zones and cessation of homecoming birth.

## Factors influencing decision-making regarding inflow and outflow for childbirth among pregnant women

In this study, we conducted a logistic regression analysis that included the number of children and maternal age as explanatory variables. The results showed that pregnant women in the 25–34 age group and primiparas were more likely to move for childbirth-related purposes. The number of children is related to the migration patterns of pregnant women because multiparous women who have school-going children may be hesitant to move out of the prefecture due to anxiety about the procedures for changing schools, psychological stress caused by the earthquake, and the possible increase of stress in an unfamiliar environment. In case of women with infants and young children who need to be taken care of by their mothers, it may be difficult for pregnant women to move around alone or unaided. Pregnant women need prenatal checkups and access to the healthcare system, which are highly important in case of complications during pregnancy. Furthermore, although the results differ from those in this study, a survey of community members who participated in the screening program at the time of the GEJE reported that those under the age of 20 or living with children tended to evacuate voluntarily [44]. Thus, the movement of pregnant women is affected not only by the scale of the disaster but also by personal and household-related factors.

In Japan, women receive support from the local government, such as childcare support, vaccinations for newborns, and health checkups. If there is a discrepancy between the residential and notification address, they may not be able to receive prompt support. Lack of administrative support is associated with postpartum depression and anxiety in pregnant women [45] and availability of such support plays an important role in promoting the health and growth of the mother and child, respectively. Therefore, pregnant women who have not received appropriate support may have experienced increased mental burden due to unresolved confusion and anxiety after the disaster. Existing studies have also reported that there is risk in giving birth in a hospital other than the one where a pregnant woman has had regular checkups [46]. Therefore, there may have been a temporary increase in high-risk deliveries in areas unaffected by the disaster.

Predicting disaster behavior in the event of a large-scale earthquake is important not just for the administration of the affected areas, to make appropriate requests for assistance, but also for those of the non-affected areas to plan their assistance for the affected areas. Japan is an earthquake-prone country and has been hit by several major earthquakes. Therefore, as a preparation for large-scale disasters in the non-affected area, during the acute phase of a disaster, the number of pregnant women who have been evacuated from the affected areas may increase, causing a temporary congestion in perinatal care. Hence, health clusters or governments need to establish a medical system or find methods to grasp the information regarding the prenatal checkups' process that can appropriately disperse evacuated pregnant women. Moreover, preparation for a major disaster in the affected area should include establishing a plan for receiving assistance during the disaster and the means to sequentially publicize the results of a multifaceted investigation on restrictions for return.

## Study strengths and limitations

The novelty of this study is that it investigates the long-term migration patterns of pregnant women after large-scale earthquakes using vital statistics that ensured complete enumeration. Furthermore, we calculated the outflow and inflow rate per 100 live births, which is a comparable outcome measure, and compared it to four large-scale earthquakes and investigated the relation between the degree of damage or other specificities and the migration patterns of pregnant women.

The study has a few limitations. First, the period between the delivery and submission of details to the birth registry affected the accuracy of data collection. In Japan, mothers are recommended to submit details to the birth registry at their place of residence to apply for child welfare and health insurance. However, the birth registry submissions can be made by family members, and other members of the household can deliver the birth registry to their residential address, which may lead to an underestimation of migration-related numbers associated with pregnant women using the registry.

Second, the model explained 60–65% of the total variance in inflow and outflow. Due to the limited information available from the vital statistics, it was not possible to add explanatory variables, and it was only possible to analyze a simple regression model. In the future, we may be able to improve the accuracy of the model further if we obtain additional information, such as how long people have been living in their place of residence, their past experiences with earthquakes, and their purpose of movement. In addition, this study revealed that the flow rate differs among prefectures in normal times. It is necessary to clarify the differences among prefectures by considering the number of medical facilities and the degree of development of public transportation systems. The Fukushima Health Management Survey conducted by the Fukushima Prefecture follows up on the pregnant women living in the affected areas in the municipal units, including the 13 evacuation ordered areas. In the future, we intend to use this data to conduct a detailed survey on a municipal unit. Furthermore, it is necessary to clarify whether there are differences in the pregnant women outflow and inflow rates by governmental restrictions on returning within the affected prefecture through a more detailed municipality analysis.

## Conclusion

After a large-scale earthquake, the migration patterns of pregnant women were found to be different from those during normal times; the outflow rate increased and inflow rate decreased. Pregnant women's migration patterns after large-scale earthquakes without the added effect of concerns regarding radiation exposure changed for just a year after the earthquake. Nevertheless, the outflow rate in Fukushima showed a significant change for 3 years post, and the inflow rate for 2 years post, suggesting that concerns for radiation exposure possibly have a significant effect. Additionally, the movement of pregnant women is related to their personal background, such as maternal age and number of children. Therefore, it is necessary to establish plans for receiving assistance and support that consider the damage caused by disasters and migration of pregnant women after disasters in both the affected and non-affected areas.

## Acknowledgments

We would like to thank Editage (www.Editage.com) for English language editing.

## Author Contributions

**Conceptualization:** Yuta Inoue, Kazutomo Ohashi, Ling Zha, Tomotaka Sobue.

**Data curation:** Yuta Inoue, Takako Fujimaki, Anna Tsutsui.

**Formal analysis:** Yuta Inoue.

**Funding acquisition:** Tomotaka Sobue.

**Methodology:** Yuta Inoue, Kazutomo Ohashi, Yuko Ohno, Tomotaka Sobue.

**Project administration:** Yuko Ohno, Tomotaka Sobue.

**Software:** Anna Tsutsui.

**Supervision:** Tomotaka Sobue.

**Writing – original draft:** Yuta Inoue.

**Writing – review & editing:** Kazutomo Ohashi, Takako Fujimaki, Anna Tsutsui.

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
