## [Decision Letter · Decision Letter 0]

4 May 2022

PONE-D-22-04835Pregnant women’s migration patterns before childbirth after a large-scale earthquake and the added impact of concerns regarding radiation exposure in Fukushima and five prefecturesPLOS ONE

Dear Dr. Inoue:

Thank you for submitting your manuscript to PLOS ONE. After careful consideration, we feel that it has merit but does not fully meet PLOS ONE’s publication criteria as it currently stands. Therefore, we invite you to submit a revised version of the manuscript that addresses the points raised during the review process.

Both reviewers suggested minor revisions that are outlined in their comments.  Please address these in a revision.

We look forward to receiving your revised manuscript.

Kind regards,

Gayle E. Woloschak, PhD

Section Editor

PLOS ONE

Journal Requirements:

Additional Editor Comments (if provided):

Both reviewers suggested minor revisions for this work. Please see the comments and address them in a revision.

Reviewers' comments:

Reviewer's Responses to Questions

**Comments to the Author**

1. Is the manuscript technically sound, and do the data support the conclusions?

Reviewer #1: Yes

Reviewer #2: Yes

2. Has the statistical analysis been performed appropriately and rigorously? 

Reviewer #1: Yes

Reviewer #2: Yes

3. Have the authors made all data underlying the findings in their manuscript fully available?

Reviewer #1: Yes

Reviewer #2: No

4. Is the manuscript presented in an intelligible fashion and written in standard English?

Reviewer #1: Yes

Reviewer #2: Yes

5. Review Comments to the Author

Reviewer #1: The paper describes the migration patterns of pregnant women after large-scale disaster in Japan. Results are well analyzed and discussion is appropriately written.

There are comments to be addressed.

1. In higher flow rates of pregnant women in Fukushima prefecture, authors referred "...concerns about radiation exposure have had long-term effects on women’s migration patterns.". How do authors manage the data of evacuees from the 13 evacuation ordered areas by the national government who needed to move out from their home town?

2. Are there any difference in outflow rate and inflow rate between evacuation area and non-evacuation area? It is very important to consider the effect of radiation concerns.

3. Authors need to correct the description in Reference as follows for example;

1) 13. Michikuni S, -> Shimo M,

2) 18. Yukiko K, -> Kobayashi Y,

3) 19. Tomoyuki S, -> Shibata T,

4) 31. Yasuhara S, -> Yasumura S,

5) 40. Ryoko F. Current reality about satogaeri bunben to support pregnant couples, choice of the birth place. Matern Health. 2018;59(2):560-568 ->Furukawa R. Current reality about satogaeri bunbeh to support pregnant couples’ choice of the birth place. Japanese Journal of Maternal Health. 2018;59(2):560-568

Reviewer #2: General comment

This is a retrospective case analyses of the migration of pregnant women after disasters using the large-scale data of governmental birth registration. Owing to the anxiety caused by the nuclear power plant accident in the 2011 Great East Japan Earthquake, I appreciate that the authors could clarify the difference between Fukushima and other prefectures. Even though Japan has a traditional trend of pregnant women going back to their maiden homes, the impact of the radiological disaster was significant.

I wonder, what will you recommend to health clusters or governments as a notion from your research in the conclusion. Mothers did not and cannot return to their original residential place not only because of fear of the radiation, but governmental restrictions for returning. There could be no conclusive scientific evaluation on which is better to go back or not. I appreciate it if you could add some notion from your discussion in conclusion on the preparedness of affected and non-affected prefectures and health clusters.

There are several points to be clarified.

Minor points

# Page 1, Line 3: Because you are dealing with various earthquakes, use "large-scale earthquakes". Same with the short title. Use "disasters".

# Page 3, Line 51: The meaning of "utilities" is not clear. I recommend to use "lifelines" instead.

# Line 66: Ref. 12 and 13 do not seem to be media reports. Add brief explanation of the notions of References 12 and 13, how the already complex radiation disaster will result in a "complex disaster".

# Line 68: I think it is better to explain how responders and governments created "emergency consultation centers for radiation exposure for pregnant women" after GEJE including their locations, duration, and the roles before citing Ref. 14. Ref. 15 is a paper on environmental exposure, but not much oriented toward radiation. Add the notion from Ref. 15.

# Line 70: It seems Ref. 18 should be in place of Ref. 19. Reorganize the references and citations.

# Line 73: Are you sure [15] is correct, not [14]?

# Line 74: It is better to use "affected area" instead of "disaster area".

# Line 77: If the data is publicly available, indicate the URL or reference for the birth registry. Indicate where the readers can find the data on the mothers’ migration. If you specifically requested the data from Ministry, describe so as written in Line 115.

# Line 79: Use "earthquakes".

# Line 81: If you are explaining the purpose of your study, "aims to clarify" instead of "suggests" may suit.

# Line 99: Please mention that the seismic resistance of houses and buildings were different by time. The experience of 1995 GHAE made the anti-seismic reinforcement in many other areas in Japan. Thus, the same seismic intensity does not destroy the same number of houses and buildings in the target earthquakes.

# Line 125: The epicenter was far out in the Pacific Ocean in GEJE. "severely affected prefecture" is fine.

# Line 133: I think the denominator is the number of births in the relevant prefecture which is the same as the denominator of outflow. "the number of births for which the place of residence was not in the relevant prefecture" can be very large. Confirm the correctness.

# Line 227: "life lines"?

# Line 231 and 237: Is Ref. 15 correct?

# Ref. 18: "Kobayashi Y" instead of "Yukiko K".

6. PLOS authors have the option to publish the peer review history of their article (what does this mean?). If published, this will include your full peer review and any attached files.

Reviewer #1: No

Reviewer #2: **Yes: **Shinichi Egawa, Professor, International Cooperation for Disaster Medicine, International Research Institute for Disaster Science (IRIDeS), Tohoku University

---

## [Author Response · Author response to Decision Letter 0]

14 Jun 2022

Thank you for allowing me to submit a revised draft of my manuscript titled “Pregnant women’s migration patterns before childbirth after a large-scale earthquake and the added impact of concerns regarding radiation exposure in Fukushima and five prefectures” to PLOS ONE. We appreciate the time and effort that you and the reviewers have dedicated to providing your valuable feedback on our manuscript. We are grateful to the reviewers for their insightful comments on our paper. We have been able to incorporate changes to reflect most of the suggestions provided by the reviewers. We have highlighted the changes within the manuscript. Here is a point-by-point response to the reviewers’ comments and concerns.

Comments from Reviewer 1

Comment 1: In higher flow rates of pregnant women in Fukushima prefecture, authors referred "...concerns about radiation exposure have had long-term effects on women’s migration patterns.". How do authors manage the data of evacuees from the 13 evacuation ordered areas by the national government who needed to move out from their home town?

Response: Thank you for pointing this out. We agree that it is important to explain how to manage the evacuation ordered areas. The background of pregnant women's migration varies: some pregnant women moved because of concerns about the effects of radiation, some pregnant women moved due to physical damage to their housing environment or medical building collapses, and some pregnant women moved because of government restrictions for returning. We have explained that calculating the rate of flow of pregnant women determines how to manage evacuation ordered areas. We have added the following sentences to explain this point at page 10, lines 151-158. 

“It is possible that the areas with government restrictions for returning are not areas where pregnant women are willing to move, but areas where they are ordered to move by the government, and the background that determined their movement is different. However, the Vital Statistics birth registry used in this study could not grasp these backgrounds. Therefore, in this study, the 13 areas ordered for evacuation were also treated as residential areas of pregnant women, without distinguishing between the intention of pregnant women and administrative restrictions on return.”

Comment 2: Are there any difference in outflow rate and inflow rate between evacuation area and non-evacuation area? It is very important to consider the effect of radiation concerns.

Response: Yes, there were. However, the change procedure of residential address after the disaster was not mandatory, and there are still some disaster victims who left their residential address in the evacuation area. Therefore, we decided that we could not treat the investigation by the municipality as an accurate result, and we reported the migration by prefecture in this study. We have added this information to the “Study strengths and limitations” sections on Page 21, lines 347-354. 

"The Fukushima Health Management Survey conducted by the Fukushima Prefecture follows up on the pregnant women living in the affected areas in the municipal units, including the 13 evacuation ordered areas. In the future, we intend to use these data to conduct a detailed survey on a municipal unit. Furthermore, it is necessary to clarify whether there are differences in the pregnant women outflow and inflow rates by governmental restrictions on returning within the affected prefecture through a more detailed municipality analysis.” 

Comment 3: Authors need to correct the description in Reference.

Response: Thank you for pointing this out. We have revised and have incorporated your suggestion throughout the reference section. We have added three new references in our revised manuscript. In addition, all the in-text citation numbers were appropriately updated. 

Comments from Reviewer 2

Comment 1: I wonder, what will you recommend to health clusters or governments as a notion from your research in the conclusion. Mothers did not and cannot return to their original residential place not only because of fear of the radiation, but governmental restrictions for returning. There could be no conclusive scientific evaluation on which is better to go back or not. I appreciate it if you could add some notion from your discussion in conclusion on the preparedness of affected and non-affected prefectures and health clusters.

Response: Thank you for this suggestion. We agree that it is important to discuss the preparedness of the prefectures and health clusters. We have added the following sentences to explain this point on Page 20, lines 316-324. 

“Therefore, as a preparation for large-scale disasters in the non-affected area, during the acute phase of a disaster, the number of pregnant women who have been evacuated from the affected area may increase, causing a temporary congestion in perinatal care. Hence, health clusters or governments need to establish a medical system or find methods to grasp the information regarding the prenatal checkups’ process that can appropriately disperse evacuated pregnant women. Moreover, preparation for a major disaster in the affected area should include establishing a plan for receiving assistance during the disaster and means to sequentially publicize the results of a multifaceted investigation of restrictions for return.”

Comment 2: Because you are dealing with various earthquakes, use "large-scale earthquakes". Same with the short title. Use "disasters".

Response: Thank you for pointing this out. We agree with this and have incorporated your suggestion.

Comment 3: The meaning of "utilities" is not clear. I recommend to use "lifelines" instead.

Response: Thank you for this suggestion. Using a dictionary, we found that public systems, such as electricity, gas, and water generally use utilities rather than lifeline. We have revised this appropriately for clarity. The revised text on Page 3, line 48 now reads as “utilities, like electricity, gas, water”.

Comment 4: Ref. 12 and 13 do not seem to be media reports. Add brief explanation of the notions of References 12 and 13, how the already complex radiation disaster will result in a "complex disaster"

Response: Thank you for pointing this out. We agree with this comment. Therefore, we have quoted the following sentence on Page 4, lines 62-65 from References 12 and 13. 

“The mechanism of health effects caused by the radiation has not yet been clarified, and opinions are divided among experts, which possibly cause anxiety not only among residents of the affected areas but also among the medical members who support them, and may even lead to harmful rumors about the affected areas.” 

Comment 5: I think it is better to explain how responders and governments created "emergency consultation centers for radiation exposure for pregnant women" after GEJE including their locations, duration, and the roles before citing Ref. 14. Ref. 15 is a paper on environmental exposure, but not much oriented toward radiation. Add the notion from Ref. 15.

Response: Thank you for pointing this out. We agree that it is important to explain the details of "emergency consultation centers for radiation exposure for pregnant women." Therefore, we have quoted the details from the Fukushima Prefecture’s website. Then, for environment exposure, we quoted the effects of environmental exposure on pregnant women from Reference 18, 19 and20. This information was added to Introduction on Page 4-5, lines 65-79.

Comment 6: It seems Ref. 18 should be in place of Ref. 19. Reorganize the references and citations.

Response: Thank you for this suggestion. We agree with this and have incorporated your suggestion. We have added three new references in the revised version of our manuscript. In addition, all in-text reference numbers were corrected and updated appropriately. 

Comment 7: Are you sure [15] is correct, not [14]?

Response: Thank you for pointing this out. We agree with this and have incorporated your suggestion.

Comment 8: It is better to use "affected area" instead of "disaster area"

Response: Thank you for pointing this out. We agree with this and have incorporated your suggestion.

Comment 9: If the data is publicly available, indicate the URL or reference for the birth registry. Indicate where the readers can find the data on the mothers’ migration. If you specifically requested the data from Ministry, describe so as written in Line 115.

Response: We agree with this and have incorporated your suggestion. We have added the sentence that we have requested the data from the Ministry on Page 5 lines 85-87.

Comment 10: Use "earthquakes"

Response: Thank you for pointing this out. We agree with this and have incorporated your suggestion.

Comment 11: If you are explaining the purpose of your study, "aims to clarify" instead of "suggests" may suit.

Response: Thank you for pointing this out. We agree with this and have incorporated your suggestion. 

Comment 12: Please mention that the seismic resistance of houses and buildings were different by time. The experience of 1995 GHAE made the anti-seismic reinforcement in many other areas in Japan. Thus, the same seismic intensity does not destroy the same number of houses and buildings in the target earthquakes.

Response: Thank you for pointing this out. We agree with this comment. Therefore, we added the sentence explaining that after disasters, the Building Standard Law was revised and the seismic strength of houses was improved. This information was added to Methods on Page 7, lines 110-113.

“Furthermore, Japan is prone to disasters such as typhoons and earthquakes and the Building Standard Law has been revised many times, each time increasing the strength of buildings. Therefore, the seismic resistance of houses and buildings were different over time.”

Comment 13: The epicenter was far out in the Pacific Ocean in GEJE. "severely affected prefecture" is fine.

Response: Thank you for pointing this out. We agree with this and have incorporated your suggestion.

Comment 14: I think the denominator is the number of births in the relevant prefecture which is the same as the denominator of outflow. "the number of births for which the place of residence was not in the relevant prefecture" can be very large. Confirm the correctness.

Response: Thank you for pointing this out. The denominator is the number of residents in one prefecture, since the outflow is the rate of pregnant women whose place of residence is in the affected prefecture and whose place of notification is in the 46 prefectures except for their place of residence. For the other, the denominator is the number of residents in 46 prefectures except for the affected prefecture, since the inflow is the rate of pregnant women whose place of notification is the affected prefecture among those whose place of residence is not the affected prefecture. Therefore, the denominator information size is different for outgoing and incoming pregnant women. We have corrected it to prevent any misunderstanding by readers on Page 9-10, lines 142-151

Comment 15: "life lines"?

Response: Thank you for this suggestion. Using a dictionary, we found that public systems, such as electricity, gas, and water generally use utilities rather than lifeline. We have revised that it to “utilities, such as electricity, gas, and water” for clarity on Page 17, line 250.

Comment 16: Is Ref. 15 correct?

Response: Thank you for pointing this out. The content refers to what happened after the GEJE, and we have decided that Reference 15 is not relevant and therefore, removed it from the reference list.

Comment 17: # Ref. 18: "Kobayashi Y" instead of "Yukiko K".

Response: Thank you for pointing this out. We revised with this and have incorporated your suggestion.

---

## [Decision Letter · Decision Letter 1]

20 Jun 2022

PONE-D-22-04835R1Pregnant women’s migration patterns before childbirth after large-scale earthquakes and the added impact of concerns regarding radiation exposure in Fukushima and five prefecturesPLOS ONE

Dear Dr. Inoue:

Thank you for submitting your manuscript to PLOS ONE. After careful consideration, we feel that it has merit but does not fully meet PLOS ONE’s publication criteria as it currently stands. Therefore, we invite you to submit a revised version of the manuscript that addresses the points raised during the review process.

We look forward to receiving your revised manuscript.

Kind regards,

Gayle E. Woloschak, PhD

Section Editor

PLOS ONE

Journal Requirements:

Additional Editor Comments:

Minor changes have been recommended.

Reviewers' comments:

Reviewer's Responses to Questions

**Comments to the Author**

1. If the authors have adequately addressed your comments raised in a previous round of review and you feel that this manuscript is now acceptable for publication, you may indicate that here to bypass the “Comments to the Author” section, enter your conflict of interest statement in the “Confidential to Editor” section, and submit your "Accept" recommendation.

Reviewer #1: (No Response)

Reviewer #2: (No Response)

2. Is the manuscript technically sound, and do the data support the conclusions?

Reviewer #1: Yes

Reviewer #2: Yes

3. Has the statistical analysis been performed appropriately and rigorously? 

Reviewer #1: Yes

Reviewer #2: Yes

4. Have the authors made all data underlying the findings in their manuscript fully available?

Reviewer #1: Yes

Reviewer #2: No

5. Is the manuscript presented in an intelligible fashion and written in standard English?

Reviewer #1: Yes

Reviewer #2: Yes

6. Review Comments to the Author

Reviewer #1: Comment 1. Please correct the authors name.

Ref. 21 Yukiko K. -> Kobayashi Y.

Ref. 22 Tomoyuki S,..., Toshimitusu H. -> Shibata T,..., Hata T.

Ref. 34 Yasuhara S,... ->Yasumura S,...

Comment 2. Is there really available of Ref. 43 (Ryoko F. ) ? The reviewer can't find the reference.

Reviewer #2: Thank you for the revision clarifying most of the concerns.

There are still several points to be clarified.

# Line 44: The Abstract section needs conclusive statement. Add what is necessary to be a resilient society summarizing the notion from Lines 319-324.

# Line 79: "fetal" or "fetuses"?

# Line 87: "applied to the Ministry for Birth registry data use" or "obtained the Birth registry data from the Ministry" may fit.

# Line 139: "that was close to the epicenter" is the correct expression.

# Line 248: Insert "Niigata Chuetsu earthquake, " between GHAE and GEJE.

# Line 260: Is "expectant pregnant women" correct? Not "expectant mothers"?

# Line 364: I think it is better to add a notion from LInes 319-324 as well as in the abstract.

7. PLOS authors have the option to publish the peer review history of their article (what does this mean?). If published, this will include your full peer review and any attached files.

Reviewer #1: No

Reviewer #2: **Yes: **Shinichi Egawa

---

## [Author Response · Author response to Decision Letter 1]

29 Jun 2022

Dear Dr. Gayle E. Woloschak,

Thank you for allowing us to submit a revised draft of our manuscript, titled “Pregnant women’s migration patterns before childbirth after a large-scale earthquake and the added impact of concerns regarding radiation exposure in Fukushima and five prefectures” to PLOS ONE. We highly appreciate your and the reviewers’ time and effort dedicated toward our manuscript as well as your valuable feedback. We are grateful to the reviewers for their insightful comments on our paper. We have incorporated the changes needed to reflect most of the suggestions provided by the reviewers. Additionally, we have highlighted the changes within the manuscript. Here is a point-by-point response to the reviewers’ comments and concerns.

Comments from Reviewer 1

Comment 1: Reviewer #1: Comment 1. Please correct the authors name.

Response: Thank you for pointing this out. We have revised the names appropriately.

Comment 2: Is there really available of Ref. 43 (Ryoko F. ) ? The reviewer can't find the reference.

Response: Yes, there is. However, it can only be confirmed on print media. This literature is written in Japanese and the author's name is Furukawa R. It has been corrected. Additionally, we have included a source link for the ease of reference. Please refer to the following URL:

https://researchmap.jp/read0112116/misc/37229744

Moreover, we have added Japanese title and the URL to the “Reference” sections on Page 27, lines 488-491.

“Furukawa R. [Current reality about satogaeri bunben to support pregnant couples’ choice of the birth place]. Syussan basyo no sentaku wo enjo suru tame no satogaeri bunben no genjou (in Japanese) [in print] Japan Society of Matern Health. 2018;59(2): 560-568. (https://researchmap.jp/read0112116/misc/37229744)”

Comments from Reviewer 2

Comment 1: # Line 44: The Abstract section needs conclusive statement. Add what is necessary to be a resilient society summarizing the notion from Lines 319-324.

Response: Thank you for this valuable suggestion. We have added the following sentences to the abstract on page 3, lines 40-43. With the addition to the abstract, it has been revised overall and is well with the journal’s stipulated limit (300 words)

“These results suggested that plans for receiving assistance and support that considers the peculiarities of disaster related damage and pregnant women’s migration patterns are needed in both the affected and non-affected areas.”

Comment 2: # Line 79: "fetal" or "fetuses"?

Response: Thank you for pointing this out. We revised this appropriately. 

Comment 3: # Line 87: "applied to the Ministry for Birth registry data use" or "obtained the Birth registry data from the Ministry" may fit.

Response: Thank you for this valuable suggestion. We have revised this appropriately for clarity by incorporating your suggestion. 

Comment 4: # Line 139: "that was close to the epicenter" is the correct expression.

Response: Thank you for pointing this out. We have revised this appropriately using your suggestion.

Comment 5: # Line 248: Insert "Niigata Chuetsu earthquake, " between GHAE and GEJE.

Response: Thank you for this valuable suggestion. We revised with this and have incorporated your suggestion.

Comment 6: # Line 260: Is "expectant pregnant women" correct? Not "expectant mothers"?

Response: Thank you for pointing this out. We have revised this term for clarity. 

Comment 7: # Line 364: I think it is better to add a notion from LInes 319-324 as well as in the abstract.

Response: Thank you for this suggestion. We have added the following sentences on Page 22, lines 364-366,

“Therefore, it is necessary to establish plans for assistance and support that consider the damage caused by disasters and migration of pregnant women after disasters in both the affected and non-affected areas.”

---

## [Decision Letter · Decision Letter 2]

18 Jul 2022

Pregnant women’s migration patterns before childbirth after large-scale earthquakes and the added impact of concerns regarding radiation exposure in Fukushima and five prefectures

PONE-D-22-04835R2

Dear Dr. Inoue:

We’re pleased to inform you that your manuscript has been judged scientifically suitable for publication and will be formally accepted for publication once it meets all outstanding technical requirements.

Kind regards,

Gayle E. Woloschak, PhD

Section Editor

PLOS ONE

Additional Editor Comments (optional):

Thank you for addressing the concerns of the reviewer.

Reviewers' comments:

Reviewer's Responses to Questions

**Comments to the Author**

1. If the authors have adequately addressed your comments raised in a previous round of review and you feel that this manuscript is now acceptable for publication, you may indicate that here to bypass the “Comments to the Author” section, enter your conflict of interest statement in the “Confidential to Editor” section, and submit your "Accept" recommendation.

Reviewer #1: All comments have been addressed

Reviewer #2: All comments have been addressed

2. Is the manuscript technically sound, and do the data support the conclusions?

Reviewer #1: Yes

Reviewer #2: Yes

3. Has the statistical analysis been performed appropriately and rigorously? 

Reviewer #1: Yes

Reviewer #2: Yes

4. Have the authors made all data underlying the findings in their manuscript fully available?

Reviewer #1: Yes

Reviewer #2: Yes

5. Is the manuscript presented in an intelligible fashion and written in standard English?

Reviewer #1: Yes

Reviewer #2: Yes

6. Review Comments to the Author

Reviewer #1: (No Response)

Reviewer #2: (No Response)

7. PLOS authors have the option to publish the peer review history of their article (what does this mean?). If published, this will include your full peer review and any attached files.

Reviewer #1: No

Reviewer #2: **Yes: **Shinichi Egawa

---

## [Editor Report · Acceptance letter]

21 Jul 2022

PONE-D-22-04835R2 

Pregnant women’s migration patterns before childbirth after large-scale earthquakes and the added impact of concerns regarding radiation exposure in Fukushima and five prefectures 

Dear Dr. Inoue:

I'm pleased to inform you that your manuscript has been deemed suitable for publication in PLOS ONE. Congratulations! Your manuscript is now with our production department. 

Kind regards, 

on behalf of

Dr. Gayle E. Woloschak 

Section Editor

PLOS ONE